# Patient Experience in Older Adults with Diabetes: A Narrative Review on Interventions to Improve Patient Experience and Research Gaps

**DOI:** 10.3390/healthcare12242530

**Published:** 2024-12-13

**Authors:** Hidetaka Hamasaki

**Affiliations:** Japan Medical Exercise Association, 4-29-16 Tsukushino, Machida, Tokyo 194-0001, Japan; h-hamasaki@umin.ac.jp; Tel.: +81-42-8505326; Fax: +81-42-8505326

**Keywords:** digital health, patient experience, patient satisfaction, randomized controlled trial, type 1 diabetes, type 2 diabetes

## Abstract

Patient experience is a critical healthcare quality indicator, evolving from Patient Satisfaction (PS) and encompassing patients’ concrete healthcare experiences. It is increasingly vital in aging societies where collaborative efforts among patients, families, and healthcare professionals are essential. Studies suggest that enhanced patient experience leads to better adherence, outcomes, and patient safety. This paper reviews patient experience evaluations in older adults with diabetes through randomized controlled trial (RCT)-based findings. The author searched PubMed/MEDLINE, Embase, AMED, and CINAHL. The review focused on RCTs examining interventions affecting patient experience and PS in T2D/T1D patients aged ≥65. A total of 13 RCTs were eligible for this review. This review highlights studies on diabetes management in older adults, assessing the impact of health education, diabetes management programs, treatments, mHealth, and advanced insulin delivery systems. Early studies showed that education improved self-care but had a limited impact on glycemic control. Key findings include the effectiveness of experience-based education in improving HbA1c, the benefits of insulin therapy for elderly patients, and the value of structured peer-to-peer diabetes management programs in enhancing satisfaction. Patient adherence, satisfaction, and personalized support emerged as critical factors influencing diabetes management across various interventions. More recent trials involving mHealth demonstrated improvements in glycemic control and PS through automated data sharing and app-based support. Closed-loop insulin delivery studies reported reduced mental strain, improved glycemic control, and better quality of life, despite barriers such as device cost and occasional system limitations. These interventions highlight the potential of advanced technologies to enhance diabetes care, particularly for aging populations. Previous RCTs show that education, structured management programs, effective insulin therapies, and advanced digital treatments improve patient experience, though well-designed studies focusing on patient experience as a primary outcome are lacking. Developing patient experience assessment scales for aging diabetes patients and adapting healthcare systems to maximize patient experience amid digitalization trends are essential, warranting further research.

## 1. Introduction

Patient experience refers to the concept of the patient’s specific “experience” related to healthcare services and is positioned internationally as an important quality indicator in healthcare, evolving from Patient Satisfaction (PS). Patient experience is defined as “events that occur to the patient either singularly or collectively throughout a series of care” [1]. In other words, patient experience refers to the patient’s concrete “experience” of healthcare services, and by its nature, the subject of evaluation is the patient. Furthermore, recently, an integrated view of human experience in healthcare, encompassing patient, workforce, and community experiences, is becoming increasingly important in improving patient experience, particularly in aging societies where patients, their families, caregivers, and healthcare professionals must collaborate to enhance health and well-being [1]. Healthcare quality consists of both Technical and Interpersonal aspects, which are complementary and interact with each other. Traditionally, PS has been used to evaluate the interpersonal relationship between physicians and patients. However, there has been criticism that the questionnaires used in PS surveys are not standardized, raising concerns about their appropriateness as objective indicators of healthcare quality. Consequently, the importance of accurately evaluating patient experience has been emphasized, and the concept has gained international recognition.

Previous studies suggest that enhanced patient experience is associated with greater adherence to recommended prevention and treatment protocols, improved clinical outcomes, better patient safety in hospitals, and reduced healthcare utilization [2]. However, caution is warranted when using patient experience measures to improve care quality, as common barriers include limited time, inadequate resources, and a lack of expertise in data analysis and quality improvement [3]. Cost-efficiency was generally low, largely due to the resources required to gather reliable samples [4]. A recent systematic review demonstrated that near-real-time feedback on patient experience, combined with data relay to providers, can enhance quality of care, even at the point of service [5]. To address issues related to resource allocation, time, and cost-efficiency in using patient experience in healthcare, advanced technologies such as wearable devices, real-time integration with electronic health records, and health data analysis with rapid, accurate diagnosis using artificial intelligence (AI) are expected to offer effective solutions (Figure 1).

Meanwhile, patients are aging in the current era. The World Health Organization projects that the global population aged 60 years and older will reach 2.1 billion by 2050 [6]. As individuals age, everyone inevitably experiences physiological changes, such as a decline in skeletal muscle mass, strength, and function. The aging process is closely linked to an increased risk of developing non-communicable diseases, including diabetes, cardiovascular diseases, and cancers [7,8]. The global incidence of diabetes continues to rise, with aging-related physiological changes driving inflammation and oxidative stress, which contribute to the development of type 2 diabetes (T2D) [9]. Additionally, the prevalence of comorbidities and vascular complications in patients with type 1 diabetes (T1D), typically diagnosed at a younger age than T2D, is increasing due to global aging [10]. As a result, the number of older adults with diabetes will continue to rise as a proportion of the total population.

Diabetes is a chronic condition, and patients often maintain lifelong relationships with healthcare providers. Given that patient experience has been shown to enhance the quality of diabetes care and potentially facilitate the achievement of treatment goals, effective bidirectional communication between patients and healthcare providers helps establish a shared understanding of treatment plans and goals, which can improve PS and clinical outcomes [11]. Specifically, patient education, management programs, treatments with high PS, and, more recently, the use of digital health technologies are believed to influence patient experience. However, as a diabetologist, the author is concerned that patient experience is not adequately assessed in daily practice, which limits opportunities to improve the quality of diabetes care. Therefore, the author investigates how these factors affect the patient experience of older adults with diabetes and whether research has provided insights to improve care quality or if there remains a research gap in this area. The author will focus on summarizing high-quality research findings from randomized controlled trials and randomized crossover trials (RCTs) and will discuss these in this paper.

## 2. Methods

This narrative review examines current English-language evidence on patient experience in older adults with diabetes. The author conducted a literature search on patient experience using PubMed/MEDLINE, Embase, Allied and Complementary Medicine Database (AMED), and Cumulative Index to Nursing and Allied Health Literature (CINAHL) from its inception to 18 November 2024. The search was limited to studies with an RCT design to focus on the effects of interventions on patient experience and PS. The inclusion criteria are as follows: (1) the mean or median age of study participants is 65 years or older; (2) all participants have T2D or T1D; and (3) the study outcomes include data on patient experience or PS. Studies investigating patient experience or PS through post hoc analyses of RCTs were excluded. The first literature search was conducted using the search terms “patient experience” and “diabetes”, resulting in a total of 340 articles. The second literature search was conducted using the search terms listed under Medical Subject Headings “patient satisfaction” and “diabetes”, resulting in a total of 691 articles. The titles and abstracts of these articles were reviewed for relevance, and the author examined the main text of each paper to determine whether the study participants’ demographics and study outcomes aligned with the inclusion criteria. Ultimately, 13 RCTs were included.

## 3. Results

The oldest study included in this review is the study by Tu et al. [12] conducted in 1993. The effectiveness of health education on the management of diabetes was assessed. A total of 27 individuals with diabetes were randomly assigned to the education group (*n* = 15) and the control group (*n* = 12). The mean age of participants was 65.6 years and 65.25 years, respectively. Over three weeks, the participants in the education group received four weekly phone calls focused on assessing and reinforcing self-care practices. Each call followed a script covering key areas: home blood glucose monitoring, medication adherence, diet, physical activity, and healthcare utilization. Where knowledge or practice gaps were identified, tailored guidance was provided, including detailed instructions for blood glucose monitoring, hypoglycemia prevention, meal timing, and exercise adjustments. Foot care was also reviewed, and referrals were made when necessary. The importance of clinic appointments and symptom reporting was highlighted to support effective diabetes management. While the education group achieved Diabetes Knowledge Scale scores 10% higher than the control group, this difference was not statistically significant. Similarly, there was no significant difference in hemoglobin A1c (HbA1c) levels between groups. However, the control group had significantly more irregularities in self-monitoring, hypoglycemia prevention, and dietary adherence. In addition, the education group showed reduced deficits in dietary adherence and symptom reporting over time. The authors suggested that follow-up after initiating health education is essential for improving self-care adherence and ensuring the safe practice of home care for older adults with diabetes.

Funnell et al. [13] examined factors that influenced attendance in an education and care program. This study was part of a 3-year RCT, the Diabetes Care for Older Adults Project. A total of 103 individuals with diabetes participated, with 53 randomly assigned to the intensive management group. The Diabetes Care for Older Adults Project was an interdisciplinary education and support initiative for older adults with T2D, led by Certified Diabetes Educators. The program’s 12-session curriculum emphasized intensive insulin therapy and self-management, including blood glucose monitoring, diet, and hypoglycemia prevention. Sessions designed for older adults, promoted peer support, included discussions of psychosocial issues, and used minimal lectures, focusing on participant interaction and practical exercises. Participants met regularly for 18 months, with additional support as needed for insulin adjustments. The program was offered without fees, and no monetary incentives were provided. Over 18 months, attendance at education and support sessions for diabetes management remained high, with 72% attendance in the first six months and 68% in the final 12 months. Participants attended an average of 7.9 sessions, citing travel, illness, and personal crises as reasons for absences. Attendance was particularly high among self-referred participants, new insulin users, and those traveling greater distances, suggesting strong motivation. Knowledge and HbA1c levels improved but were not correlated with attendance. Most participants valued the education and reported few drawbacks, with 90% willing to join similar future programs.

The study by Sarkadi et al. [14] is an RCT evaluating a 12-month, experience-based group educational program for diabetes self-management. Led by trained pharmacists and initially supported by a diabetes nurse specialist, the intervention aimed to improve glycemic control by improving body awareness and participants’ ability to estimate blood glucose levels. Educational tools included videos, interactive games, and booklets using relatable metaphors and scenarios to facilitate learning. Pharmacists provided continuous support but refrained from altering participants’ medical regimens. Analysis revealed significant short-term improvements in HbA1c levels within the intervention group at six months compared to the control group, which showed no change. However, no significant differences were observed at 12 months. At 24 months, HbA1c reductions in the intervention group remained significant. Regression models indicated initial HbA1c and satisfaction with diabetes-specific knowledge were key predictors of outcomes, with the latter mediating the intervention’s effects. Duration and treatment variations did not significantly impact results. Initial HbA1c and participation in the intervention program significantly influenced HbA1c change. Key differentiators included diabetes knowledge satisfaction and exercising for glucose control. This study highlights the potential of experience-based education to improve long-term glycemic control, emphasizing knowledge satisfaction as a critical factor in diabetes management outcomes.

Hendra et al. [15] investigated whether well-being status changed after administrating insulin therapy. Fifty-seven elderly T2D patients were recruited from clinics. Participants were assessed for vascular complications, cognitive function (Abbreviated Mental Test, Folstein Minimental State Examination), and daily living ability (Barthel Index). Subjects were randomized into three treatment groups: continuation of oral medication (Group 1), twice-daily isophane insulin (Group 2), or basal-bolus insulin (Group 3). Over six months, HbA1c, weight, hypoglycemia, glucose profiles, and health-related outcomes were monitored. Health status was assessed using the Short Form Health Survey (SF-36), and treatment satisfaction was evaluated through the Diabetes Treatment Satisfaction Change Questionnaire (DTSQ). Mood was measured using the Hospital Anxiety and Depression Scale. Carers completed the General Health Questionnaire to assess psychological distress. Subjects received diabetes education and glucose monitoring training, with insulin titration overseen by a diabetes nurse. At baseline, groups were similar in demographics, diabetes duration, cognitive function, and BMI, though Groups 1 and 3 had fewer vascular complications. Over six months, HbA1c levels significantly improved in Groups 1 and 3 but not in Group 2. Group 3 showed significant health status improvements across multiple SF-36 domains, including vitality, social function, and mental health. Weight gain was observed in insulin-treated groups. Anxiety scores decreased significantly in Groups 1 and 3, while depression scores improved only in Group 3. Carer burden remained stable across groups. Overall, basal–bolus insulin (Group 3) demonstrated superior glycemic and psychological outcomes compared to other treatments, supporting its effectiveness in older adults with T2D.

There is an RCT examining whether practice nurses can manage people with T2D as safely as general practitioners [16]. Participants were recruited from a group practice with five general practitioners. Of 230 randomized patients, 206 (102 intervention, 104 control) completed the 14-month follow-up. Patients in the intervention group were managed by two practice nurses trained for one week on diabetes care protocols based on Dutch guidelines. The practice nurses were authorized to prescribe or adjust medications, order laboratory tests, and manage diabetes-related care except for initiating insulin therapy. The control group received standard general practitioner care. The primary outcome was the change in HbA1c levels. Secondary outcomes included blood pressure, cholesterol levels, target achievement for glycemic and cardiovascular indicators, and measures for eye and foot care. Both groups showed significant reductions in systolic and diastolic blood pressure, but no significant differences were observed between groups for HbA1c or lipid profiles. The practice nurse group referred more patients for ophthalmology and foot care. Patients in the intervention group experienced a decline in health-related QoL and an increase in diabetes-related symptoms. However, those treated by practice nurses reported higher satisfaction with their care compared to patients under the care of general practitioners.

The second Diabetes Glucose and Myocardial Infarction trial evaluated glucose-lowering strategies in 1253 patients with T2D and suspected acute myocardial infarction [17]. Patients were randomized to one of three approaches: (1) acute insulin–glucose infusion followed by long-term multi-dose insulin; (2) insulin–glucose infusion followed by standard therapy; or (3) standard glucose-lowering treatment per local practice. Over a median follow-up of 2.1 years, no significant differences in glucose control, cardiovascular events, or mortality were observed among the groups. A subset of 533 patients from Nordic countries participated in a QoL sub-study, with 324 completing assessments at baseline and 12 months. Patients were categorized based on discharge treatments: insulin or oral glucose-lowering agents. QoL was evaluated using three measures: DTSQ, Psychological General Well-Being Index, and Rating Scale. Treatment satisfaction and psychological well-being improved significantly over 12 months in both groups, with no major differences between them. Insulin-treated patients reported a lower perceived health Rating Scale at baseline and follow-up but experienced comparable improvements over time. Patients achieving better glycemic control (lower HbA1c) reported greater QoL improvements across all measures. Self-reported hypoglycemia was more frequent in the insulin group and correlated with lower treatment satisfaction. Men reported higher treatment satisfaction and psychological well-being than women at baseline, but both sexes showed significant improvements. Changes in BMI and transitions between insulin and oral treatments had minimal impact on QoL. Overall, sustained glycemic control and reduced hypoglycemic events were critical to improving QoL outcomes.

Müller et al. [18] investigated the importance of the injection-to-meal interval (IMI) in people with T2D undergoing flexible insulin therapy using human insulin. This prospective, randomized, open-label, single-center crossover trial included 100 participants who completed two 12-week study phases: one with a 20 min IMI and the other without. Primary and secondary outcomes assessed glycemic control (HbA1c), incidence of hypoglycemia, insulin dose, QoL, treatment satisfaction, and patient preference. Participants monitored blood glucose levels and recorded insulin usage during the trial. HbA1c slightly increased without the IMI compared to with it, which was deemed clinically insignificant, confirming non-inferiority. Mild hypoglycemia incidents slightly decreased without the IMI, though the difference was not statistically significant. No cases of severe hypoglycemia were reported. Insulin doses and blood glucose profiles did not differ significantly between phases. Psychosocial evaluations showed a marked improvement in treatment satisfaction without the IMI (average increase: 8.08 points), while overall QoL remained unchanged. In the study’s conclusion, 86.5% of participants preferred insulin therapy without an IMI, irrespective of the trial phase they concluded with. These findings suggest that omitting the IMI does not compromise metabolic control and improves PS, indicating it may be a viable option for people with T2D using regular human insulin.

Valentiner et al. [19] explored the feasibility and effectiveness of using electronic momentary assessments, goal-setting, and personalized phone calls to improve adherence to a 12-week self-guided interval walking training (IWT) program, which was delivered via the InterWalk smartphone app, among people with T2D. Participants were randomly assigned to either the experimental group (IWT with additional support) or the control group (IWT without support). Both groups used the InterWalk app to perform at least three IWT sessions per week. The experimental group received additional support, including automated text messages, electronic momentary assessments, and personalized phone calls if they reported non-adherence. The primary outcome was adherence to IWT, measured by total minutes of IWT over the 12 weeks. Secondary outcomes included physical activity levels, health-related QoL, aerobic capacity, and changes in HbA1c and anthropometric measures. These outcomes were derived from a qualitative investigation. Results showed that the experimental group accumulated significantly more IWT minutes compared to the control group. Forty-seven percent of the experimental group reached the recommended 90 min per week of IWT, compared to 11% in the control group. The experimental group also had higher satisfaction rates, with 68% of participants very satisfied with the trial. No significant between-group differences were found in metabolic outcomes, though mental well-being improved in the experimental group. This study suggests that the additional support in the experimental group improved adherence to IWT.

The Spanish researchers evaluated the effects of a diabetes self-management program on patient experience [20]. This study evaluated a structured diabetes self-management program compared to usual care for people with T2D. Participants were randomly assigned to intervention or control groups after baseline measurements, with data collected at 6, 12, and 24 months. The intervention involved weekly 2.5 h workshops over six weeks, facilitated by a healthcare professional and a person with diabetes or a caregiver. Usual care consisted of primary care and individual diabetes education by nurses. The primary outcome was HbA1c levels, while secondary outcomes included cardiovascular factors, medication use, QoL, self-efficacy, physical activity, dietary habits, PS, and healthcare resource use. A total of 594 participants were enrolled, with 80% attending at least four sessions. Over two years, no significant differences in HbA1c or cardiovascular outcomes were observed between groups. However, participants in the intervention group used fewer medications and had fewer nurse visits, particularly in the first year. Self-efficacy scores improved significantly in the intervention group, along with a tendency for better QoL. Satisfaction was high among intervention participants, with most reporting improved knowledge, disease control, and lifestyle changes. While physical activity improved in both groups, no differences were noted between them. These findings suggest that a diabetes self-management program benefits self-efficacy and patient experience but shows limited impact on clinical outcomes.

Recent studies, perhaps reflecting current trends, often focus on patient experience with the use of advanced medical devices and IoT technologies.

Reznik et al. [21] investigated the safety and effectiveness of an automated insulin delivery (AID) system for people with T2D who struggled to independently manage multiple daily insulin injections (MDI) at home. Thirty participants were randomized into either a control group (continued MDI) or an intervention group (AID system using the Tandem tX2 pump and continuous glucose monitoring (CGM) system, Dexcom G6). AID users underwent training, system calibration, and monitoring by visiting nurses and healthcare providers. Primary outcomes included time in the target glucose range (TIR), while secondary outcomes encompassed HbA1c, insulin dose, weight, and QoL. The intervention group demonstrated a significant TIR improvement of 27.4% compared to the control group. HbA1c reductions were greater in the AID group (1.61% vs. 0.36%), with 90.9% achieving HbA1c below 8% by study’s end. Safety was comparable, with no severe hypoglycemic events reported. Device deficiencies were more frequent in the intervention group but did not impact outcomes. PS with the AID system was high, with 77.8% expressing confidence and willingness to continue. Overall, the AID system improved glycemic control and was well received by participants and healthcare providers, indicating its potential as an effective alternative for T2D management.

Sun et al. [22] reported that an mHealth system is effective for managing older adults with T2D. PS was evaluated six months after the intervention using a custom questionnaire. In this RCT, 91 individuals were randomly assigned to either the intervention group (*n* = 44) or the control group (*n* = 47). Patients in the intervention group were trained to use an mHealth app to upload their glucometer data, which were automatically transmitted to a medical server. Every two weeks, the medical team reviewed the data and provided advice through messaging or phone. The control group used conventional outpatient follow-ups. Both groups received dietary and exercise guidance, with the intervention group receiving app-based support and the control group receiving face-to-face counseling. Regular follow-ups, including physical exams and blood tests, were conducted every three months for both groups. Both the control and intervention groups showed significant improvements in HbA1c levels at three months, with no difference between groups. However, the intervention group had significantly lower HbA1c levels than the control group at six months (6.84% vs. 7.22%) and demonstrated greater overall reductions. Additionally, patients in the intervention group experienced a significant decline in postprandial blood glucose levels. Satisfaction surveys revealed an average score of 6.3/7, indicating improvements in self-monitoring, dietary management, and diabetes knowledge. A systematic review has also shown that telemedicine, including eHealth and mHealth, is highly accepted by people with diabetes [23]; however, comprehensive patient experience was not evaluated in previous studies.

Recently, the lived experiences of older adults with long-duration T1D who began using a closed-loop automated insulin delivery system for the first time were investigated [24]. The authors assessed the patient experience by semi-structured interview. This study revealed five major themes. First, participants reported differences between their expectations and the reality of closed-loop therapy. Though initial expectations of effortless glucose control were high, users acknowledged some continued need for input, particularly around blood glucose testing and carbohydrate entry, but still appreciated the significant reduction in diabetes management effort compared to traditional methods. In terms of glucose control, many participants experienced improved stability, especially overnight, which also eased their fear of hypoglycemia. The system’s automatic responses to glucose fluctuations reduced their psychological burden, allowing more flexibility with diet and routine. Family members shared this relief, trusting the system to handle hypoglycemic events. QoL improved, notably with better sleep quality as nighttime glucose levels remained steady. However, while participants gained trust in the system over time, alarms—especially at night—were a source of frustration, particularly for calibration. Lastly, cost was a significant barrier, as many struggled with the financial burden of CGMs and insulin pumps, sometimes delaying retirement or sacrificing other expenses to afford the technology. Overall, while closed-loop therapy did not fully eliminate the daily management of diabetes, it provided notable benefits in glucose stability, flexibility, and reduced mental strain.

Similarly, Schneider-Utaka et al. [25] reported that a hybrid closed-loop insulin delivery system improved patient experience in older adults with T1D. A total of 37 participants were randomly assigned to the CamAPS FX hybrid closed-loop system group (*n* = 20) and sensor-augmented therapy group (*n* = 17). After the 4-month intervention period, QoL, diabetes distress, glucose monitoring satisfaction, hypoglycemic confidence, and satisfaction with the device were measured. Of the study participants, 19 completed user experience interviews. Nearly all (95%) indicated they would continue using the CamAPS FX system if available, wishing they had access to it earlier to prevent diabetes complications. Many (63%) reported reduced mental and emotional burden, with one participant stating it felt “the closest to normal I’ve felt in those 51 years”. Despite positive feedback, some users found the pump challenging, describing it as difficult to use, small, and time-consuming. Additionally, several experienced hypoglycemia during moderate to intense exercise, which they attributed to the system’s algorithm.

The characteristics of the included studies are summarized in Table 1. The findings of the studies included in this review reveal that those evaluating patient experience, including PS, employed various intervention methods. These methods comprised educational and support programs, workshops, insulin therapy (e.g., timing of insulin injections), patient management by practice nurses, advanced insulin therapies, and mHealth technologies. Across these interventions, glycemic control was either improved or remained stable. While almost all studies reported improvements in PS and reductions in diabetes-related distress, changes in patients’ QoL were inconsistent, with some studies showing no change or even a decline. Given the substantial heterogeneity in intervention approaches, drawing definitive conclusions about which methods most effectively enhance patient experience remains challenging. Nevertheless, advanced diabetes treatments, such as those incorporating mHealth and insulin pumps, consistently appear to improve PS, treatment-related knowledge, adherence, and QoL.

## 4. Discussion

This review demonstrated that educational programs improved diabetes knowledge, satisfaction with diabetes-specific knowledge, and attendance at care appointments. Achieving good glycemic control and reducing hypoglycemic events through the implementation of insulin therapy enhanced treatment satisfaction and psychological well-being. However, people with diabetes seem dissatisfied with complex and troublesome insulin injection methods, such as IMI. Interestingly, people with diabetes were more satisfied with their care by practice nurses than general practitioners. The diabetes self-management program benefits self-efficacy and patient experience. Digital technologies, such as mHealth apps, have been shown to improve glycemic control, psychosocial well-being, treatment adherence, and self-efficacy in diabetes management. These benefits may be further amplified through personalized support. Furthermore, advanced diabetes treatment technologies, such as hybrid closed-loop insulin delivery systems, were effective in enhancing patient experience with self-management and reducing diabetes-related distress.

### 4.1. Research Gaps

Evidence shows a positive relationship between patient experience, PS, and clinical effectiveness, consistent across various diseases, study designs, and patient populations. Across studies, improved PE correlates with better clinical outcomes like lower mortality, improved blood glucose control, reduced infections, and fewer medical errors, as well as with improved self-reported health and well-being outcomes, including better QoL and reduced anxiety. Adherence to treatment recommendations and the use of preventive care were also positively associated with patient experience. For instance, studies show that strong physician–patient communication significantly improves adherence to treatment, increasing compliance by 1.62 times when physicians have communication training. Patients receiving effective communication and developing good relationships with healthcare providers exhibit higher medication adherence and better engagement in preventive services, such as diabetes and cancer screenings [26]. However, most of the included studies were cohort or cross-sectional studies, with only 4 out of 55 being RCTs [26]. The present review identifies a significant research gap, noting that very few RCTs have investigated the effects of certain interventions on patient experience as primary outcomes. To clarify the specifics of how, when, where, and for whom patient experience enhances care quality in older adults with chronic conditions like diabetes, more evidence from RCTs is needed.

Moreover, the studies included reported patient-reported outcome measures (PROMs) rather than patient-reported experience measures (PREMs). PREMs differ from PROMs, as they focus on evaluating patients’ experiences with care rather than health status, and from PS metrics, which indicate the extent to which a patient’s expectations were met. However, patient experience is often critiqued for being heavily influenced by prior healthcare encounters [27]. Developing patient-reported indicators now emphasizes co-development and co-design, where patients, healthcare providers, patients’ families, and caregivers collaborate as partners in creating, implementing, and evaluating such measures. This approach has become the standard for PROMs and PREMs, aiming to ensure that measures reflect patient priorities and improve the quality of care [28]. The Consumer Assessment of Healthcare Providers and Systems Clinician and Group (CG-CAHPS) survey measure is one of such reliable and valid indicators and can be used for quality improvement and applied for interventions aiming at the improvement of patient-centered care [29,30]. However, when selecting PREMs for research and evaluation, it is crucial to consider additional criteria, such as whether a disease- or setting-specific measure is more suitable than a generic one, and whether a PREM developed in the researcher’s country is more applicable than one created abroad, potentially tailored to a different healthcare system [27]. Thus, the author suggests that patient experience measurement instruments should align with specific age groups, as patient experience ratings vary significantly across age groups, with older adults (over 65 years) consistently reporting higher satisfaction than younger adults (18–34 years) [31]. Diabetes is a lifelong disease, making continuity of care and the interpersonal relationship between physician and patient crucial. Researchers should employ appropriate patient experience measurement instruments when evaluating the effects of interventions on patient experience in patients with diabetes [32,33]. The development of tools that accurately assess patient experience specifically for older adults with diabetes is anticipated.

In this review, 4 out of 13 studies were about advanced technology for diabetes treatment. A growing number of studies have been conducted on the effectiveness of digital health on patient experience recently. Wang et al. [34] proposed the definition of digital patient experience as follows: “Digital patient experience is the sum of all interactions affected by a patient’s behavioral determinants, framed by digital technologies, and shaped by organizational culture, that influence patient perceptions across the continuum of care channeling digital health”. As digitalization in healthcare rapidly advances, encompassing vital sign monitoring via wearable devices like smartphones and smartwatches, as well as blood glucose management through CGM systems leveraging IoT technology, assessing patient experiences of digitalization has become an indispensable concern. Diabetes, in particular, is a condition in which continuous blood glucose monitoring and maintaining glucose fluctuations within physiological ranges are crucial for preventing complications and improving prognosis [35,36,37]. Therefore, it is a disease where digital health can play a substantial role. On the other hand, interestingly, it has been reported that diabetes patients tend to prefer face-to-face consultations more than patients without diabetes [38]. In an aging society, telemedicine can be employed to reduce the burden on patients requiring care, such as those with advanced diabetes complications and declining activities of daily living, while still providing in-person consultations as needed to address their psychological and social care needs. Striking the best balance between digital and in-person care is essential, as is striving for optimal integration in healthcare.

### 4.2. Study Limitations

This narrative review has several limitations. Firstly, it was conducted by a single author, which may introduce biases in study selection and assessment. To address this, future systematic reviews should involve multiple authors to independently conduct literature searches and evaluations. Secondly, in this review, only one qualitative study met the inclusion criteria. Individual qualitative analyses provide highly valuable insights into research on patient experience and PS, highlighting the need for further studies focusing on older adults with diabetes. Thirdly, it is important to note that this narrative review is limited to studies focusing on older adults with diabetes aged 65 or older, and does not discuss patient experience in the general diabetic population. Furthermore, it does not comprehensively investigate the effects of various interventions such as enhancing communication, patient-centered approaches, supporting self-management, and promoting multidisciplinary collaboration, which have been effective in improving patient experience in previous reports [39,40]. Future efforts should focus on accumulating evidence and deepening our understanding of interventions that impact patient experience in older adults with diabetes. Finally, the concept of patient experience has evolved significantly in recent years, broadening its scope, which means that findings from older studies may no longer align with current perspectives. Initially driven by consumer movements, patient experience has undergone shifts in the role of patients within clinical settings, the introduction of national measurement tools, and the establishment of dedicated leadership roles. It is now recognized as an independent dimension of healthcare quality. Historically, healthcare transitioned from being viewed as a service to an experience industry in the 1980s, and the establishment of Planetree in 1978 advanced human-centered care; patient experience has shifted from risk management to a culture of intentionally designed experiences [41]. In addition, the rise of digital health has been pivotal in advancing patient-centered care, with patient engagement solutions becoming central to patient experience research. AI and digital technologies are enhancing patient understanding, trust, and involvement, with AI-driven chatbots offering 24/7 support and personalized health assessments. By prioritizing individual needs and values, patient-centered care encourages active participation, leading to improved outcomes. Digital solutions are transforming both patient experience and care delivery, reshaping healthcare systems through innovation and a shift in mindset [42].

## 5. Conclusions

In conclusion, previous RCTs have shown that structured health education and management programs, insulin therapy achieving stable glycemic control, and treatments using advanced digital technologies are effective in improving patient experience. However, a significant research gap remains due to the lack of well-designed intervention studies that establish patient experience as a primary outcome. Considering the aging of the patient population and the specific characteristics of diabetes—namely, the need for lifelong treatment and the impact of the physician–patient interpersonal relationship on both patient experience and treatment outcomes, along with the psychosocial barriers, such as diabetes distress, often faced by people with diabetes—it may be necessary to develop patient experience assessment scales tailored to older adults with diabetes. In addition, as digital healthcare rapidly advances, it is essential to consider approaches to healthcare that maximize patient experience. If patient experience is conceptualized as encompassing not only the patient but also family members, healthcare providers, caregivers, and the broader social environment, then the ways in which digitalization can enhance patient experience, improve care quality, and increase its value within healthcare may also shift to align with societal digitalization trends. Further research in this area is highly expected.

## Figures and Tables

**Figure 1 healthcare-12-02530-f001:**
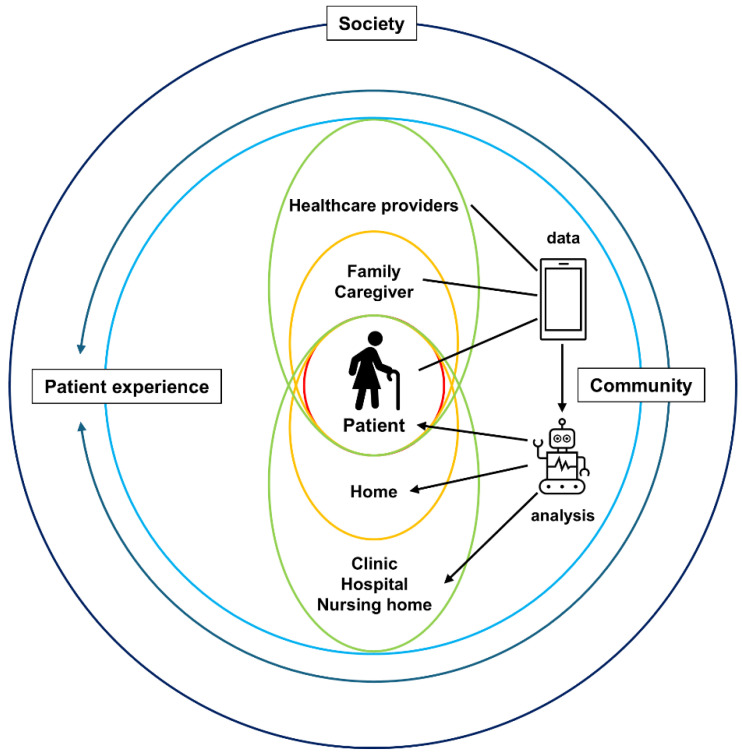
Patient experience is a holistic concept encompassing patients, healthcare professionals, caregivers, as well as families, healthcare institutions, and care facilities. It is shaped by interactions with patients and can be regarded as a community issue that is crucial for enhancing healthcare quality in an aging society. Moreover, recent advancements in IoT technology have made real-time connections between patients and healthcare providers or facilities possible. With the progress of digital healthcare, patient experience is expected to evolve further.

**Table 1 healthcare-12-02530-t001:** Summary of included studies.

Reference	Country	Study Design	Study Period	Subjects	Study Outcomes	Intervention	Results
Tu et al. (1993) [12]	USA	Randomized, controlled, parallel-group trial	4 weeks	27 older adults with diabetes, Intervention group 5 men and 10 women): Age: 65.6 ± 7.0 years, BMI: No description, HbA1c: 11.76 ± 3.1% Control group (4 men and 8 women): Age: 65.25 ± 6.00 years, BMI: No description, HbA1c: 11.33 ± 1.67%	Diabetes self-care knowledge HbA1c	Health education program	Adherence ↑ Knowledge → HbA1c →
Funnell et al. (1998) [13]	USA	Randomized, controlled, parallel-group trial	18 months	103 older adults with diabetes, 77% of whom were on insulin therapy, Intervention group (24 men and 29 women): Age: 69.5 ± 4.3 years, BMI: No description, HbA1c: 12.5 ± 2.4% Control group (*n* = 50, sex ratio unknown): Age: No description, BMI: No description, HbA1c: No description	Factors associated with the program attendance	Education and support program	Factors: insulin use duration, distance Knowledge ↑ HbA1c ↓
Sarkadi et al. (2004) [14]	Sweden	Randomized, controlled, parallel-group trial	12 months	64 older adults with type 2 diabetes, Intervention group (*n* = 33, sex ratio unknown): Age: 66.4 ± 7.9 years, BMI: 27.2 ± 3.6 kg/m^2^, HbA1c: approximately 6.5% Control group (*n* = 31, sex ratio unknown): Age: 66.5 ± 10.7 years, BMI: 28.6 ± 5.8 kg/m^2^, HbA1c: approximately 6.5%	Primary: HbA1c Secondary: patient satisfaction with diabetes knowledge	Experience-based group educational program	HbA1c ↓
Hendra et al. (2004) [15]	UK	Randomized, 3-arm, parallel-group trial	6 months	51 older adults with type 2 diabetes, Group 1 (9 men and 10 women): Age: 71.4 ± 5.9 years, BMI: No description, HbA1c: 9.6 ± 1.7% Group 2 (9 men and 10 women): Age: 69.4 ± 5.4 years, BMI: No description, HbA1c: 10.0 ± 1.8% Group 3 (9 men and 10 women): Age: 68.8 ± 7.4 years, BMI: No description, HbA1c: 9.4 ± 1.7%	Health status, mood, diabetes treatment satisfaction, HbA1c	Insulin therapy	Health status ↑, depression scores ↓ HbA1c →
Houweling et al. (2011) [16]	Netherlands	Randomized, controlled, parallel-group trial	14 months	206 older adults with type 2 diabetes, Intervention group (54 men and 48 women): Age: 67.1 ± 11.0 years, BMI: 30.6 ± 5.3 kg/m^2^, HbA1c: 7.6 ± 1.3% Control group (44 men and 60 women): Age: 69.5 ± 10.6 years, BMI: 30.3 ± 4.5 kg/m^2^, HbA1c: 7.4 ± 1.3%	Primary: HbA1c Secondary: blood pressure, lipid profile, patient satisfaction, health-related QoL, diabetes-related symptoms, health care spending	Practice nurses	Patient satisfaction ↑, health-related QoL ↓ HbA1c →, blood pressure →, lipid profile →
Venskutonyte et al. (2013) [17]	Sweden	Randomized, 2-arm, parallel-group trial	12 months	324 older adults with type 2 diabetes and acute myocardial infarction, Insulin therapy group (98 men and 77 women): Age (median): 67 years, BMI (median): 28 kg/m^2^, HbA1c (median): 7.6% Oral treatment group (85 men and 42 women): Age (median): 67 years, BMI (median): 28 kg/m^2^, HbA1c (median): 6.7%	Patient satisfaction, psychological well-being	Insulin therapy	Patient satisfaction →, psychological well-being →
Müller et al. (2013) [18]	Germany	Randomized, crossover trial	24 weeks	100 older adults with type 2 diabetes, Injection-to-meal interval first group (23 men and 26 women): Age: 66.9 ± 7.5 years, BMI: 34.5 ± 6.6 kg/m^2^, HbA1c: 7.1 ± 0.7% Injection-to-meal interval last group (25 men and 26 women): Age: 66.6 ± 7.5 years, BMI: 33.9 ± 5.2 kg/m^2^, HbA1c: 7.0 ± 1.7%	Primary: HbA1c Secondary: mild and severe hypoglycemia, mean blood glucose, diabetes treatment satisfaction, QoL, patient preference for the injection-to-meal interval	Injection-to-meal interval	HbA1c → Hypoglycemia →, treatment satisfaction ↑, QoL →
Valentiner et al. (2019) [19]	Denmark	Randomized, 2-arm, pilot, parallel-group trial	12 weeks	37 older adults with type 2 diabetes, Experimental group (6 men and 13 women): Age: 66.7 ± 7.3 years, BMI: 29.0 ± 6.0 kg/m^2^, HbA1c: 49.1 ± 11.6 mmol/mol Control group (7 men and 11 women): Age: 65.1 ± 6.4 years, BMI: 29.8 ± 5.6 kg/m^2^, HbA1c: 51.6 ± 12.9 mmol/mol	Primary: adherence to interval walking training Secondary: physical activity levels, health-related QoL, aerobic capacity, glycemic control, anthropometric measures (qualitative analysis)	InterWalk app + prompts, ecological momentary assessment, interviews, and personalized phone calls	Adherence ↑ Participation satisfaction ↑
Gamboa Moreno et al. (2019) [20]	Spain	Randomized, controlled, parallel-group trial	24 months	594 older adults with type 2 diabetes, Intervention group (191 men and 106 women): Age (median): 65 years, BMI: 30.4 ± 4.8 kg/m^2^, HbA1c: 7.2 ± 1.3% Control group (164 men and 133 women): Age (median): 65 years, BMI: 30.2 ± 5.1 kg/m^2^, HbA1c: 7.1 ± 1.2%	Primary: HbA1c Secondary: cardiovascular-related factors, medication use, QoL, self-efficacy, dietary habit, physical activity levels, patient satisfaction, the number of visits to the healthcare institutions	6 weekly structured peer-to-peer workshops	HbA1c → Cardiovascular-related factors →, self-efficacy ↑, patient satisfaction ↑, medication consumption ↓, healthcare use rates ↓
Reznik et al. (2024) [21]	France	Randomized, controlled, parallel-group trial	12 weeks	29 older adults with type 2 diabetes, Intervention group (7 men and 7 women): Age: 69.3 ± 6.7 years, BMI: 32.3 ± 4.3 kg/m^2^, HbA1c: 9.0 ± 1.2% Control group (2 men and 13 women): Age: 69.7 ± 10.3 years, BMI: 35.6 ± 6.5 kg/m^2^, HbA1c: 9.25 ± 1.0%	Primary: the percentage of time in the target glucose range of 70–180 mg/dL Secondary: HbA1c, total daily insulin dose, body weight and BMI changes, patient and healthcare provider experience	Automated insulin delivery	The percentage of time in the target glucose range of 70–180 mg/dL ↑ HbA1c ↓, patient and health care provider satisfaction ↑
Sun et al. (2019) [22]	China	Randomized, controlled, parallel-group trial	6 months	91 older adults with type 2 diabetes Intervention group (19 men and 25 women): Age: 67.9 years, BMI: 23.3 kg/m^2^, HbA1c: 7.84 ± 0.73% Control group (18 men and 29 women): Age: 68.04 years, BMI: 23.6 kg/m^2^, HbA1c: 7.88 ± 0.64%	Primary: HbA1c, blood glucose levels Secondary: patient satisfaction	mHealth app 2024	HbA1c ↓ Patient satisfaction ↑ Knowledge ↑ Self-management efficacy ↑
Kubilay et al. (2023) [24]	Australia	Randomized, crossover trial	4 months	21 older adults with type 1 diabetes (14 men and 7 women): Age: 67 ± 4 years, BMI: No description, HbA1c: No description	Patient experience	Closed-loop insulin delivery system vs. sensor-augmented pump therapy	QoL ↑ Glycemic control ↑ Barriers: usability, cost
Schneider-Utaka et al. (2023) [25]	UK	Randomized, parallel-group trial	4 months	37 older adults with type 1 diabetes (21 men and 16 women): Age: 67 ± 5 years, BMI: No description, HbA1c: 7.4 ± 0.9%	Patient experience	Hybrid closed-loop insulin delivery system vs. sensor-augmented pump therapy	Diabetes distress ↓ Glycemic control ↑ Barriers: usability, time-consuming, device size

↑ increase or improve, → no change, ↓ decrease, BMI body mass index, HbA1c hemoglobin A1c, QoL quality of life.

## Data Availability

The data that support the findings of this study are available from the corresponding author upon reasonable request.

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
