# Peer review of "Patient Experience in Older Adults with Diabetes: A Narrative Review on Interventions to Improve Patient Experience and Research Gaps"

_healthcare, 2024, doi:10.3390/healthcare12242530_

Round 1
Reviewer 1 Report (Previous Reviewer 1)
Comments and Suggestions for Authors
Thanks for the response and the revision of your manuscript. An interesting and important paper,

Author Response
It is an interesting paper and important result. Some further revision may enhance the
understanding of the paper.
Thank you very much for taking the time to review the revised manuscript again. I have addressed your comments and suggestions as outlined below. I would greatly appreciate it if you could review them.
Abstract
Is it ok with abbreviations in abstract according to journal?
According to the journal's instructions for authors, “Acronyms/Abbreviations/Initialisms should be defined the first time they appear in each of three sections: the abstract, the main text, and the first figure or table.” Therefore, the use of abbreviations in the abstract is acceptable.
Introduction
You use abbreviations Patient Experience (PEX), for the readability it is easier without.
Thank you for your suggestion. The term "patient experience" has now been written out in full.
Method
The first literature search was conducted using the search terms “patient experience” and
“diabetes,” resulting in a total of 340 articles. The second literature search was conducted using the
search terms 114 “patient satisfaction” and “diabetes,” resulting in a total of 691 articles. This is not
clear, why not block searching and how many duplicates in these articles.
“Patient satisfaction” is listed under Medical Subject Headings (MeSH), whereas “patient experience” is not. Therefore, the author conducted two separate literature searches based on these terms. Duplications between the two searches were not accounted for.
Result
Your result does not seem to be synthesized at all, Your result present article for article, Similarity
or differences?
In response to your comment, the author has added the following text to the end of the Results section:
“The findings of the studies included in this review reveal that those evaluating patient experience, including PS, employed various intervention methods. These methods comprised educational and support programs, workshops, insulin therapy (e.g., timing of insulin injections), patient management by practice nurses, advanced insulin therapies, and mHealth technologies. Across these interventions, glycemic control was either improved or remained stable. While almost all studies reported improvements in PS and reductions in diabetes-related distress, changes in patients’ QoL were inconsistent, with some studies showing no change or even a decline. Given the substantial heterogeneity in intervention approaches, drawing definitive conclusions about which methods most effectively enhance patient experience remains challenging. Nevertheless, advanced diabetes treatments, such as those incorporating mHealth and insulin pumps, consistently appear to improve PS, treatment-related knowledge, adherence, and QoL.”
Discussion
The discussion refers to your result and limitation, consider a title for limitations as well as for your conclusion, now it is presented all together.
The following subheadings have been added to the Discussion section: Research Gaps, Study Limitations, and Conclusion.
Reviewer 2 Report (Previous Reviewer 2)
Comments and Suggestions for Authors
The author responded to the reviewers’ criticisms in detail and reflected them in the article. Substantial improvements were made to sections such as the title, abstract, methodology, and discussion. This shows that the author is open to feedback and is willing to improve his work. Thank you very much for this. The new title reflects the focus of the study more clearly. The reviewers’ comments regarding the shortcomings of the methodology were taken into account and methodological information was added to the abstract. The methodological limitations of the study (e.g., single-author study, limited qualitative data) were clearly stated and supported with suggestions for future research. The literature search was extended to PubMed/MEDLINE as well as to a wider database such as Embase, CINAHL, and AMED. The rationale for including only RCTs was clearly explained and aligned with the purpose of the study. The revisions were generally acceptable and show that the author meticulously addressed the criticisms.
Author Response
The author responded to the reviewers’ criticisms in detail and reflected them in the article. Substantial improvements were made to sections such as the title, abstract, methodology, and discussion. This shows that the author is open to feedback and is willing to improve his work. Thank you very much for this. The new title reflects the focus of the study more clearly. The reviewers’ comments regarding the shortcomings of the methodology were taken into account and methodological information was added to the abstract. The methodological limitations of the study (e.g., single-author study, limited qualitative data) were clearly stated and supported with suggestions for future research. The literature search was extended to PubMed/MEDLINE as well as to a wider database such as Embase, CINAHL, and AMED. The rationale for including only RCTs was clearly explained and aligned with the purpose of the study. The revisions were generally acceptable and show that the author meticulously addressed the criticisms.
Thank you for taking the time to review the manuscript. Your comments and suggestions have greatly helped the author improve the article.
This manuscript is a resubmission of an earlier submission. The following is a list of the peer review reports and author responses from that submission.
Round 1
Reviewer 1 Report
Comments and Suggestions for Authors
Hi,
Thanks for interesting paper and important aspects for older adults with diabetes. However there is some needs and suggestions for clarifications in this paper.

Author Response
Review- Patient Experience in Older Adults with Diabetes: A Review
Interesting paper to review and important aspects according to older adults with diabetes, however there is a need of clarifications, according to aim, title, methodology and limitations.
Thank you for taking the time and effort to review my manuscript. I will respond to your comments and suggestions point by point as follows.
Title and abstract
Suggest a title according to the aim,
The title was changed to “Patient Experience in Older Adults with Diabetes: A Narrative Review on Interventions to Improve Patient Experience and Research Gaps.”
Methods needs to be provided in abstract.
The revised abstract includes a concise description of the search methodology.
Keywords normally in alphabetic order, suggest revision
The keywords have been reordered in alphabetical order.
Introduction
Background/rationale
The scientific background has been reported, the problem formulation needs some clarifications what is the problem, why is it a problem, what is your mission. It is expressed -This review highlights studies on diabetes management in older adults, assessing the impact of health education, mHealth, and advanced insulin delivery systems. Early studies showed education improved self-care but had limited impact on glycemic control. The background does not refer to these concepts.
Thank you for your helpful comments. In accordance with your comments, the author has revised the last paragraph of the Introduction section as follows:
“Diabetes is a chronic condition, and patients often maintain lifelong relationships with healthcare providers. Given that PEX has been shown to enhance the quality of diabetes care and potentially facilitate the achievement of treatment goals, effective bidirectional communication between patients and healthcare providers helps estab-lish a shared understanding of treatment plans and goals, which can improve PS and clinical outcomes [11]. Specifically, patient education, management programs, treatments with high PS, and, more recently, the use of digital health technologies are believed to influence PEX. However, as a diabetologist, the author is concerned that PEX is not adequately assessed in daily practice, which limits opportunities to improve the quality of diabetes care. Therefore, the author investigates how these factors affect the PEX of older adults with diabetes and whether research has provided insights to improve care quality or if there remains a research gap in this area.”
Objectives
the author investigates how the PX of older adults with diabetes is being evaluated and whether research findings have been gained to improve the quality of care in the end of the background, In abstract you refer to other objective, this need to be clear.
In abstract you express - This review highlights studies on diabetes management in older adults, assessing the impact of health education, mHealth, and advanced insulin delivery systems
Thank you for your careful reading. The author has revised the manuscript to ensure consistency between the abstract and the introduction. I would appreciate it if you could check the revised version.
Methods
Study design
Data collection
Data analysis
This narrative review is expressed, according to which reference? The method needs to express data collection procedures clearer, and dataanalysis should be replicable It is hard to follow the procedure, design, datacollection with appropriate procedures and dataanalysis need to be developed. No tables are presenting your datacollection, as for example PRISMA or search strategy.
Thank you for your comment. Narrative reviews do not require strict adherence to PRISMA guidelines, as PRISMA is primarily designed for systematic reviews and meta-analyses. However, the author has revised the Methods section to improve readership as follows:
“This narrative review examines current English-language evidence on PEX in older adults with diabetes. The author conducted a literature search on PEX using Pub-Med/MEDLINE, Embase, Allied and Complementary Medicine Database (AMED), and Cumulative Index to Nursing and Allied Health Literature (CINAHL) from its inception to 18 November 2024. The search was limited to studies with an RCT design to focus on the effects of interventions on PEX and PS. The inclusion criteria are as follows: (1) the mean or median age of study participants is 65 years or older; (2) all participants have T2D or T1D; and (3) the study outcomes include data on PEX or PS. Studies investigating PEX or PS through post hoc analyses of RCTs were excluded. The first literature search was conducted using the search terms “patient experience” and “diabetes,” resulting in a total of 340 articles. The second literature search was conducted using the search terms “patient satisfaction” and “diabetes,” resulting in a total of 691 articles. The titles and abstracts of these articles were reviewed for relevance, and the author examined the main text of each paper to determine whether the study participants' demographics and study outcomes aligned with the inclusion criteria. Ultimately, 13 RCTs were included.”
Results The result represent the effectiveness of health education on the management of diabetes , not represented in your aim or presented in title or background. Please revise. No synthesized result,
The author has revised the Introduction section to clearly present the aim of this review based on the current evidence. The synthesized results are also described in the first paragraph of the Discussion.
Discussion
Key results
You have Summarised your key results
Limitations
No methodological limitations are expressed or the limitation in a review as the only author, ethical aspects ?
The study's limitations have been added to the Discussion section as follows:
“This narrative review has several limitations. Firstly, it was conducted by a single author, which may introduce biases in study selection and assessment. To address this, future systematic reviews should involve multiple authors to independently conduct literature searches and evaluations. Secondly, in this review, only one qualitative study met the inclusion criteria. Individual qualitative analyses provide highly valuable insights in research on PEX and PS, highlighting the need for further studies focusing on older adults with diabetes.”
Generalisability
No discussion of transferability or generalizability
Regarding the generalizability of this review, the author has added the following text to the Discussion section:
“Thirdly, it is important to note that this narrative review is limited to studies focusing on older adults with diabetes aged 65 or older, and does not discuss PEX in the general diabetic population. Furthermore, it does not comprehensively investigate the effects of various interventions such as enhancing communication, patient-centered approaches, supporting self-management, and promoting multidisciplinary collaboration those are effective in improving PEX in previous reports [39,40]. Future efforts should focus on accumulating evidence and deepening our understanding of interventions that influence PEX in older adults with diabetes.”
Concluding remarks
Language
OK
References OK
Reviewer 2 Report
Comments and Suggestions for Authors
Is it the abbreviation for Patient Experience (PX)?
It is abbreviated as PX throughout the text. It should be corrected
Only PubMed/MEDLINE was used. A broader database coverage, such as CINAHL, Scopus or Web of Science, would have allowed for a more comprehensive literature review.
Including only RCT studies excluded a larger group of studies that assessed patient experiences (e.g. observational studies, qualitative studies). The reason should be explained
The inclusion of only 5 studies from the initial set of 320 and 355 articles indicates a rather narrow scope. The reasons for including such a small number of studies and the detailed criteria for these exclusions (e.g. what the specific inclusion/exclusion criteria were) should be reported more clearly.
The search was conducted between "inception to October 2024", but this broad time period did not take into account the possibility that the concept of PX may have changed over time. For example, the concept of PX has gained broader meanings in recent years and the findings of older studies may not be consistent with the present. This should be explained in detail.
Having a single researcher perform the search, screening and data analysis processes may cause subjective evaluations to come into play in the process. In particular, the risk of bias increases in the review of titles and abstracts or in the interpretation of findings in studies. In an ideal system, the literature review and screening process should be conducted independently by at least two researchers. Such a double-blind approach increases the accuracy of the study and allows the selected articles to be evaluated from a broader perspective. Therefore, it is not appropriate for a single researcher to manage the process.
Standards such as PRISMA (Preferred Reporting Items for Systematic Reviews and Meta-Analyses) ensure that processes in literature reviews are transparent and reproducible. Such standards generally recommend that more than one researcher be involved in the process. Having a single researcher in this study may have caused the process to be overlooked. PRISMA should be explained.
It should be considered whether there is a PROSPERO registration.
The reasons why 5 out of 355 remain should be explained. The reasons for elimination should be determined.
The method is not appropriate.
Author Response
Is it the abbreviation for Patient Experience (PX)?
It is abbreviated as PX throughout the text. It should be corrected
Thank you for taking the time to review the manuscript. I would appreciate it if you could confirm that the revised manuscript has been appropriately updated.
The abbreviation for patient experience has been changed from PX to PEX.
Only PubMed/MEDLINE was used. A broader database coverage, such as CINAHL, Scopus or Web of Science, would have allowed for a more comprehensive literature review.
The author has expanded the database search to include PubMed/MEDLINE, Embase, AMED, and CINAHL, resulting in a total of 13 eligible studies being included in this review.
Including only RCT studies excluded a larger group of studies that assessed patient experiences (e.g. observational studies, qualitative studies). The reason should be explained
Thank you for your comments. While associations between patient experience and various factors have been identified, causal relationships cannot be established from the results of observational studies. This review focuses on investigating the effects of interventions on patient experience and satisfaction. Therefore, only RCTs, which are capable of assessing the impact of interventions, were included.
Additionally, one RCT that incorporated a qualitative analysis of patient satisfaction was included. Two other seemingly eligible qualitative studies were identified: “Type 2 diabetic patients’ experiences of two different educational approaches—A qualitative study” [Int J Nurs Stud. 2008;45(7):986-994] and “Implementing an Advance Care Planning Intervention in Community Settings with Older Latinos: A Feasibility Study” [J Palliat Med. 2017;20(9):984-993]. However, the former study did not meet the inclusion criteria as participants were under 65 years old, and the latter included patients with various other conditions (e.g., cancer, heart disease, renal/liver failure, stroke, hypertension, chronic obstructive pulmonary disease, and HIV/AIDS) in addition to diabetes.
The inclusion of only 5 studies from the initial set of 320 and 355 articles indicates a rather narrow scope. The reasons for including such a small number of studies and the detailed criteria for these exclusions (e.g. what the specific inclusion/exclusion criteria were) should be reported more clearly.
According to your comments, the author has revised the Methods as follows:
“This narrative review examines current English-language evidence on PEX in older adults with diabetes. The author conducted a literature search on PEX using Pub-Med/MEDLINE, Embase, Allied and Complementary Medicine Database (AMED), and Cumulative Index to Nursing and Allied Health Literature (CINAHL) from its inception to 18 November 2024. The search was limited to studies with an RCT design to focus on the effects of interventions on PEX and PS. The inclusion criteria are as follows: (1) the mean or median age of study participants is 65 years or older; (2) all participants have T2D or T1D; and (3) the study outcomes include data on PEX or PS. Studies investigating PEX or PS through post hoc analyses of RCTs were excluded. The first literature search was conducted using the search terms “patient experience” and “diabetes,” resulting in a total of 340 articles. The second literature search was conducted using the search terms “patient satisfaction” and “diabetes,” resulting in a total of 691 articles. The titles and abstracts of these articles were reviewed for relevance, and the author examined the main text of each paper to determine whether the study participants' demographics and study outcomes aligned with the inclusion criteria. Ultimately, 13 RCTs were included.”
The search was conducted between "inception to October 2024", but this broad time period did not take into account the possibility that the concept of PX may have changed over time. For example, the concept of PX has gained broader meanings in recent years and the findings of older studies may not be consistent with the present. This should be explained in detail.
Thank you for your suggestion. The author discussed this aspect of patient experience in the literature in the study limitation part as follows: “Finally, the concept of PEX has evolved significantly in recent years, broadening its scope, which means that findings from older studies may no longer align with current perspectives. Initially driven by consumer movements, PEX has undergone shifts in the role of patients within clinical settings, the introduction of national measurement tools, and the establishment of dedicated leadership roles. It is now recognized as an independent dimension of healthcare quality. Historically, healthcare transitioned from being viewed as a service to an experience industry in the 1980s, and the establishment of Planetree in 1978 advanced human-centered care. PEX has shifted from risk management to a culture of intentionally designed experiences [41]. In addition, the rise of digital health has been pivotal in advancing patient-centered care, with patient engagement solutions becoming central to PEX research. AI and digital technologies are enhancing patient understanding, trust, and involvement, with AI-driven chatbots offering 24/7 support and personalized health assessments. By prioritizing individual needs and values, patient-centered care encourages active participation, leading to improved outcomes. Digital solutions are transforming both PEX and care delivery, reshaping healthcare systems through innovation and a shift in mindset [42].”
Having a single researcher perform the search, screening and data analysis processes may cause subjective evaluations to come into play in the process. In particular, the risk of bias increases in the review of titles and abstracts or in the interpretation of findings in studies. In an ideal system, the literature review and screening process should be conducted independently by at least two researchers. Such a double-blind approach increases the accuracy of the study and allows the selected articles to be evaluated from a broader perspective. Therefore, it is not appropriate for a single researcher to manage the process.
Standards such as PRISMA (Preferred Reporting Items for Systematic Reviews and Meta-Analyses) ensure that processes in literature reviews are transparent and reproducible. Such standards generally recommend that more than one researcher be involved in the process. Having a single researcher in this study may have caused the process to be overlooked. PRISMA should be explained.
The author agrees with your comment. However, the present review is a narrative review, not a systematic review. Narrative reviews do not require strict adherence to PRISMA guidelines, as PRISMA is intended for systematic reviews and meta-analyses. Nevertheless, the author has revised the Methods section to enhance readability, as noted above, and believes there is no need to follow PRISMA guidelines for this review.
Additionally, the author has acknowledged in the Discussion section that one of the major limitations of this review is the risk of bias due to study selection being conducted by a single author. The text states: “This narrative review has several limitations. Firstly, it was conducted by a single author, which may introduce biases in study selection and assessment. To address this, future systematic reviews should involve multiple authors to independently conduct literature searches and evaluations.”
It should be considered whether there is a PROSPERO registration.
A narrative review is not required to be registered in PROSPERO. PROSPERO is primarily a registration system for systematically planned reviews, such as systematic reviews and meta-analyses. In contrast, narrative reviews do not employ systematic search or analytical methods but instead synthesise literature based on the researcher’s insights and interpretations, making registration unnecessary. Therefore, the author believes that PROSPERO registration is not required for this review.
The reasons why 5 out of 355 remain should be explained. The reasons for elimination should be determined. The method is not appropriate.
As explained in the Methods section, the author reviewed the main text of each paper to determine whether the study participants' demographics and outcomes met the inclusion criteria. A total of 13 RCTs were included in the revised manuscript.